# The Hemostatic System in Newborns and the Risk of Neonatal Thrombosis

**DOI:** 10.3390/ijms241813864

**Published:** 2023-09-08

**Authors:** Jamilya Khizroeva, Alexander Makatsariya, Alexander Vorobev, Victoria Bitsadze, Ismail Elalamy, Arina Lazarchuk, Polina Salnikova, Sabina Einullaeva, Antonina Solopova, Maria Tretykova, Alexandra Antonova, Tamara Mashkova, Kristina Grigoreva, Margaret Kvaratskheliia, Fidan Yakubova, Natalia Degtyareva, Valentina Tsibizova, Nilufar Gashimova, David Blbulyan

**Affiliations:** 1Department of Obstetrics, Gynecology and Perinatal Medicine, N.F. Filatov Clinical Institute of Children’s Health, I.M. Sechenov First Moscow State Medical University (Sechenov University), Trubetskaya Str. 8-2, 119991 Moscow, Russia; gemostasis@mail.ru (A.M.); alvorobev@gmail.com (A.V.); vikabits@mail.ru (V.B.); ismail.elalamy@aphp.fr (I.E.); arina.lazarchuk@mail.ru (A.L.); salnikovapolina@bk.ru (P.S.); eynullayevas00@gmail.com (S.E.); antoninasolopova@yandex.ru (A.S.); tretyakova777@yandex.ru (M.T.); antonova.snk@inbox.ru (A.A.); iza.62@mail.ru (T.M.); grigkristik96@gmail.com (K.G.); margaret.kv@mail.ru (M.K.); fi_dan_2017@mail.ru (F.Y.); soba4ka-10@yandex.ru (N.D.); nelya.94@yandex.ru (N.G.); blbulyan@gmail.com (D.B.); 2Hematology and Thrombosis Center, Tenon Hospital, Sorbonne University, 4 Rue de la Chine, 75020 Paris, France; 3Almazov National Medical Research Centre, Health Ministry of Russian Federation, 2 Akkuratova Str., 197341 Saint Petersburg, Russia; tsibizova.v@gmail.com

**Keywords:** neonatal hemostasis, neonatal thrombosis, risk factors for neonatal thrombosis, COVID-associated thrombosis, LMWH, anticoagulant therapy

## Abstract

Newborns are the most vulnerable patients for thrombosis development among all children, with critically ill and premature infants being in the highest risk group. The upward trend in the rate of neonatal thrombosis could be attributed to progress in the treatment of severe neonatal conditions and the increased survival in premature babies. There are physiological differences in the hemostatic system between neonates and adults. Neonates differ in concentrations and rate of synthesis of most coagulation factors, turnover rates, the ability to regulate thrombin and plasmin, and in greater variability compared to adults. Natural inhibitors of coagulation (protein C, protein S, antithrombin, heparin cofactor II) and vitamin K-dependent coagulation factors (factors II, VII, IX, X) are low, but factor VIII and von Willebrand factor are elevated. Newborns have decreased fibrinolytic activity. In the healthy neonate, the balance is maintained but appears more easily converted into thrombosis. Neonatal hemostasis has less buffer capacity, and almost 95% of thrombosis is provoked. Different triggering risk factors are responsible for thrombosis in neonates, but the most important risk factors for thrombosis are central catheters, fluid fluctuations, liver dysfunction, and septic and inflammatory conditions. Low-molecular-weight heparins are the agents of choice for anticoagulation.

## 1. Introduction

Neonatal thrombosis is a rare (0.7–0.14 per 10,000) but dangerous condition, leading to significant morbidity and mortality among newborns [1]. From 1997 to 2018, the number of newborns with neonatal thrombosis increased by 13 times [1,2]. About 2% to 4% of neonates die due to venous thromboembolism (VTE) [3,4]. The risk of neonatal thrombosis increases after the admission of a child to the neonatal intensive care unit (NICU) and the overall incidence of thrombosis in NICU patients is about 0.2% [5]. According to a prospective 2-year registry of VTE in children in the Netherlands, 85% of patients develop thrombosis during the patient’s stay in the hospital [3]. Two children out of ninety-nine died as result of VTE.

The clinical presentation can be nonspecific or even asymptomatic, which is why the number of thrombosis cases in neonates is underestimated [4,6].

One of the essential pathogenetic factors is the condition of the coagulation system in newborns, which is different from that of the older children and adults [1]. Coagulation proteins do not cross the placenta, but are synthesized in the fetus from an early stage. At birth, activities of the vitamin K-dependent factors II, VII, IX, and X and the concentrations of the contact factors XI and XII are decreased to about 50% of normal adult values and approach adult level by 6 months of life [7]. Conversely, levels of natural anticoagulants (antithrombin, heparin cofactor II and protein C and S) are low at birth. The fibrinolysis system is characterized by reduced level of plasminogen and alpha-2-antiplasmin, an increased tissue plasminogen activator. These features all tend to be gestational dependent, and therefore are more present in the preterm infant. Despite these features of neonatal hemostasis, the healthy newborn rarely develops spontaneous VTE but appears more easily vulnerable to thrombosis. Different triggering factors are responsible for thrombosis in neonates.

Additional research is needed to better characterize the biology of the developing hemostasis system in normal and critically ill children, to identify clinically relevant criteria and tests for evaluation, as well as to determine effective treatment strategies.

## 2. Neonatal Hemostasis System Peculiarities and Neonatal Thrombosis

### 2.1. Epidemiology of Thrombosis in Neonates

According to studies conducted in Canada, Germany, and Denmark, the incidence of thrombosis among newborns is approximately 2.4/1000 children in ICU and 5.5/100,000 births in general. The meta-analysis summarizing the incidence of thrombosis in ICU children found that the incidence rate of ICU thrombosis is approximately 2%, which is consistent with previous studies [8].

The first Canadian registry of neonates with thrombosis was established in 1990 and included data from 15 tertiary-care pediatric centers [9]. One hundred and thirty-seven patients with thrombosis were identified. The overall incidence of DVT/PE was 5.3/10,000 hospital admissions or 0.07/10,000 children in this registry.

A case registry at McMaster University included data from physicians in 64 centers in North America, Europe and Australia [10]. Ninety-seven cases of venous thromboembolism (VTE) were registered. Twenty-one neonates developed renal vein thrombosis and thirty-nine children had other localizations of VTE. In 89% of the cases, thrombosis development was associated with catheter placement, but sepsis and extensive surgery were also the most important risk factors.

A study conducted in Germany reported the incidence of VTE as 5.1 per 100,000 births, with a total of 79 cases of neonatal thrombosis registered [11]. The diagnosis was confirmed by Doppler ultrasonography. Renal vein thrombosis occurred in 35 neonates, with another VTE localization in 25, and arterial thrombosis developed in other cases. Thrombosis associated with various risk factors developed in 59 cases: central venous catheter placement (*n* = 25), asphyxia (*n* = 13), septicemia (*n* = 11), dehydration (*n* = 6), maternal diabetes (*n* = 2), and heart disease (*n* = 1). Genetic thrombophilia was diagnosed in seven cases.

The Denmark study from 1994 to 2006 included patients from 0 to 18 years old diagnosed with VTE and found age- and sex-related disparities in the incidence of pediatric venous and arterial thrombosis [12]. The highest incidence was registered in children under 1 year, especially males. Risk factors were presented in 86.6% of cases, and 47.9% of newborns were diagnosed with inherited thrombophilia.

The report from the Italian Registry of Pediatric Thrombosis represented 75 neonates (0–28 days) with thromboembolism. The data were collected from neonatology centers from 2007 to 2013. VTE was observed in 41 (55%) neonates, arterial thrombosis in 22 (29%), and other participants had cerebral venous thrombosis. A total of 65% of children were male and in 29 (25%) cases thrombosis was diagnosed on the first day of life [6]. A total of 70% of cases were associated with maternal/placental risk factors in the early-onset group, and 33% of patients were diagnosed with inherited thrombophilia. Postnatal risk factors were associated with catheter use and infection in 73% of all cases. The study also suggested that corticosteroid use in preterm infants is an additional risk factor. Systemic glucocorticoids exert prothrombotic effects via the reducing the clearance of activated clotting factors and through direct vasoconstriction. However, the association between thrombosis development and glucocorticoid use is still inexplicit.

The incidence of stroke in neonates and premature neonates is 25 per 100,000 population per year. Half of these are ischemic [1]. In North America, the incidence of neonatal ischemic stroke was estimated to be 2.5 to 2.7 cases per 100,000 children per year, and in France it was 3 cases per 100,000 children per year. Ischemic stroke is among the top 10 causes of death among children in the United States, with the highest rate during the first year of life [7].

The incidence of cerebral venous sinus thrombosis is approximately 0.4 per 1000 newborns.

### 2.2. Hemostasis System in the Neonate

#### 2.2.1. Neonatal Hemostatic Balance

The fetal/neonatal hemostatic system differs from the adult one. The classical clotting pathway undergoes a series of chemical reactions through two pathways. The first pathway is extrinsic and starts with the activation of tissue factor. The second cascade is an intrinsic pathway that begins with the contact activation of factor XII. Fetus coagulation factors do not cross the placenta, and the first synthesis of them is appears in the 11th week of gestational age. At weeks 19–27 there are still low levels of many factors, except the von Willebrand factor (VWF). At 28–31 weeks of pregnancy, VWF and fibrinogen levels are normal but other factors are still low. Most components of the coagulation system in neonates achieve an adult level by 6 months (Table 1) [13,14].

At first glance, neonates have a tendency to bleed, due to specific physiological mechanisms. However, their bleeding time and the time it takes to form a clot is paradoxically less than in adults [15,16,17]. Despite the prolongation of APTT, PT and delayed thrombin generation, a healthy newborn is not more prone to bleeding or thrombosis. There is an evolving “hemostatic balance” of pro-and anticoagulant factors due to which a healthy newborn hardly ever develops VTE (Figure 1) [14]. Despite these age-related differences between coagulation factors, the production of thrombin in newborns is equivalent to about 90% of that in adults, which is enough for the formation of a hemostatic clot [18].

Physiologically low levels of inhibitors could compensate for low concentrations of clotting factors and ensure sufficient thrombin production, but the results obtained with standard clotting tests do not confirm this assumption: it has been shown that the ability to produce thrombin in the plasma of a healthy newborn is markedly reduced and delayed compared to an adult. Only 30–50% of peak thrombin activity can be produced in neonatal plasma compared to adults.

Cvirn et al. demonstrated that the simultaneous effect of low anticoagulant capacity of the three inhibitors (activated protein C (APC), tissue factor pathway inhibitor (TFPI) and antithrombin) leads to a reduction in clotting time and faster formation of factor (F) Xa and thrombin in the umbilical cord compared to adult plasma when small amounts of TF (<10 microns) are used to initiate clotting [19]. This can explain the clinically observed excellent hemostasis of neonates despite low levels of procoagulant factors [20].

In most healthy newborns, infants, and young children, normal hemostasis mechanisms can compensate for differences in pro- and anticoagulant factors, preventing serious bleeding and thrombotic complications. However, neonatal hemostasis has less buffer capacity and can be easily tipped over into thrombosis by acquired risk factors such as concomitant diseases, decreased fibrinolytic capacity, heparin resistance due to low antithrombin (AT), higher clearance of unfractionated heparin (UFH), and increased sensitivity to anticoagulants. The upward trend in the rate of neonatal thrombosis could be attributed to progress in the treatment of severe neonatal conditions and increased survival in the premature babies. The compensatory mechanisms of premature neonates are less developed. Premature babies have even bigger differences of pro- and anticoagulant factors due to lower levels of clotting factors, but they accelerate fast towards “normal”.

In 1987, Andrew et al. published the first reference values for PT and activation time of blood platelets among healthy term babies [13]. In 1988, a study was conducted on healthy babies born within 30 to 36 weeks [21]. Studies mentioned above included children aged 1, 5, 30, or 90 days old. The postnatal maturation of clotting factors to the level of an adult occurs at an accelerated rate in premature infants compared to full-term infants. By 6 months, most of the components of the blood clotting system in premature infants reach values close to those of adults [21]. In the premature birth protein C, the mean value is 38%; protein S is even lower (26–28%), and is not compensated by a2-macroglobulin (110%)—which, however, starts rising quickly. The activity of vitamin K-dependent coagulation factors is particularly reduced in preterm infants as compared to term-born infants. The majority of premature neonates have increased platelet-VWF interaction and are more prone to sepsis. The bleeding time of preterm neonates born after 37 weeks’ gestation is approximately two times longer than those born earlier (*p* < 0.001) [15].

In the largest Irish cross-sectional study of very premature infants (<30 weeks gestational age) there was no difference in the potential of endogenous thrombin between the plasma of premature and full-term infants. Despite the prolongation of clotting time, thrombin generation was similar in very preterm and full-term neonates [22]. Neary et al. showed that in preterm children, levels of coagulation proteins II, VII, and IX and protein S, as well as antithrombin, are lower than in term children.

The role of heparin cofactor II (so called “minor” antithrombin) in the modulation of neonatal hemostasis system is still questioned. HCII selectively inhibits thrombin and has a moderate binding affinity to heparin and dermatan sulfate. The rate of thrombin inhibition by HCII is significantly slower that by antithrombin, and the plasma level of HCII is 25–50% that of AT. Term infants and healthy preterm newborns have a lower level of HCII, which more likely reflects immature liver function, and they attain adult levels at 5 to 7 months of life. Neither adult or children with a thrombosis history had significantly lower levels of HCII activity. Considering that the absence of HCII does not strongly correlate with thrombosis, its physiological functions are still under investigation.

The level of plasma soluble thrombomodulin, which reflects endothelial damage, is increased at birth in asphyxiated full-term infants.

Thrombomodulin (TM) is a multidomain glycoprotein receptor for thrombin and is best known for its role as a cofactor in a clinically important natural anticoagulant pathway of protein C. It binds thrombin and converts it from a procoagulant to an anticoagulant enzyme that activates protein C. TM is expressed at the surface of endothelial cells and exists in a soluble form (sTM) in plasma and urine. Loss of TM from the surface of endothelial cells may favor thrombosis. TM not only has anticoagulant activity but also exerts anti-inflammatory properties using APC-dependent and APC-independent mechanisms. In addition to its anticoagulant and anti-inflammatory function, the TM–thrombin complex activates thrombin-activatable fibrinolysis inhibitor (TAFI), which inhibits tissue plasminogen activator (tPA)-induced fibrinolysis (antifibrinolytic properties of TM). Inflammation has been shown to reduce TM expression on the endothelial surface and this decrease may contribute to the hypercoagulable condition which is characteristic of inflammatory states. Plasma thrombomodulin concentration and plasma thrombomodulin-to-serum creatinine ratio at birth were even higher in very-low-birthweight infants than those in full-term infants [23].

#### 2.2.2. Platelet Count and Function in the Neonate

Platelet function and physiology also depend on age. The results of the largest study on the hemostasis system formation in neonates has demonstrated that the platelet count in the fetus during pregnancy increases by ~2 × 10^9^/L for each week of gestation. Even in premature infants, the average platelet count was ≥200 × 10^9^/L (within the normal range for an adult) [24]. The study showed that the postpartum period had a significant effect on the number of platelets; during the first 9 weeks, the indicators corresponded to a sinusoidal pattern with two peaks: one at 2–3 weeks, and the second at 6–7 weeks. The upper limit of the expected values (95th percentile) during these peaks reached 750,000 per microliter of blood.

The average platelet count in children is similar to that in adults. The normal range of platelet counts in newborns and infants is from 150 × 10^3^ to 450 × 10^3^/mcL, although some data indicate a slightly lower limit for normal, especially in premature infants [25]. Preterm neonates may have a higher incidence of thrombocytopenia and bleeding, most commonly in the brain. Nevertheless, the function of the blood cells shows the great differences.

Platelet activation in vitro demonstrates decreased activation and response to various inducers, e.g., collagen, ADP, thromboxane A_2_ (TxA_2_), thrombin, and epinephrine. Neonatal platelet reactivity increases with gestational age, demonstrating that platelet reactivity is age-dependent. In preterm neonates, platelet hyporeactivity occurs due to decreased membrane glycoprotein expression [26]. Platelets collected from infants of less than 30 weeks of gestation expressed lower levels of membrane glycoproteins (GP), exposed less P-selectin on their surface [27], and were less reactive than platelets from term newborns [28]. In the same study, a significantly lower level of glycoprotein (GPIIb/IIIa) expression on platelets from peripheral blood was seen in term newborns as well as preterm infants, compared to adults.

It is now clear that platelets perform not only hemostatic but also important non-hemostatic functions, especially in angiogenesis, immune responses and inflammation [29,30].

The umbilical cord blood platelets of full-term and premature newborns showed a reduced response to most physiological agonists. This hyporeactivity is partly due to both insufficient synthesis and reaction to an important mediator of platelet function—thromboxane A_2_ (TxA_2_). The poor response of newborn platelets to TxA_2_ is not due to differences in the characteristics of binding to the TxA_2_ receptor compared to platelets of the control group of adults. It was reported the post-receptor signal transduction pathway through the TxA_2_ receptor was affected in newborns, and this defect in signal transduction through phospholipase C-β (PLCβ) contributes to the observed poor response of newborns’ platelets to TxA_2_ and consequently to TxA_2_-dependent agonists such as collagen. This post-receptor defect in signal transduction in cord blood platelets may explain different abnormalities in platelet function of newborn infants, including poor aggregation and secretion responses, decreased PLC activity, impaired calcium mobilization, decreased thromboxane production, and decreased response to an important mediator of platelet function TxA_2_ [31].

Neonatal platelets are also hyporesponsive to the tyrosine kinase-linked receptor agonist collagen, which is related to a reduced expression of glycoprotein VI (GPVI) and C-type lectin-like receptor 2 (CLEC-2) [32].

Decreased platelet function in newborns is associated with different factors:Platelet hyporeactivity to epinephrine is caused by a reduced number of alpha-2 (α2) adrenergic receptors on the cell surface.Decreased thrombin platelet response occurs due to the lack of protease-activated receptor 1 (PAR-1) and PAR-4 receptors on the neonatal platelets [33].Reduced signal transduction results in thromboxane hyporeactivity in the newborn.Reduced platelet activation after collagen inducing is associated with a lack of GPVI receptors, combined with defects in intracellular signaling pathways. Evidence may be seen in an insufficient phosphorylation of Syk and CLEC-2 in neonatal platelets [32].

A recent study showed a neonate’s platelets’ special features compared to an adult. Platelets of a newborn display hypersensitivity while inhibited by the prostaglandin E1 (PGE1). The blood cells also show increased PGE1-induced cAMP levels. However, the biological significance of this mechanism is still not clear, and the tests show the increased functioning of the PGE2-ADP-Proteinkinase II axis [34].

Platelet alpha granules (α-granules) contain clotting system growth factors and proteins, e.g., von Willebrand factor, P-selectin, coagulation factors, and others. Caparros et al. demonstrated that SNARE family proteins like Stx11 and its regulator Munc18b and β1-tubulin have reduced secretion. This phenomenon is associated with decreased α-degranulation in platelets. However, a decreased β1-tubulin level has no effect on platelet morphology [35].

### 2.3. Clinical Features of Neonatal Thrombosis

The new Italian Registry of Infantile Thrombosis (RITI) is the largest available European registry of neonatal and pediatric thrombosis, and includes a total of 2668 questions. The RITI collected the data of 1001 neonates and children affected by cerebral or systemic thrombosis from 48 Italian pediatric and intensive care units. Available data showed that 57.8% of affected neonatal and pediatric patients were male; the age at first thrombotic event was median 0.9 years; 24.8% had neonatal cerebral thrombosis, 8.7% had neonatal systemic thrombosis, 41.5% had pediatric cerebral thrombosis and 25.1% had pediatric systemic thrombosis [36]. Attention is drawn to the high frequency of cerebral thrombosis.

Perinatal arterial ischemic stroke (PAIS) is a cerebrovascular disorder which includes a group of arterial ischemic injuries that can affect full-term and premature infants in the prenatal, perinatal and postpartum periods. Different types of perinatal arterial ischemic stroke have different clinical manifestations, risk factors, and long-term outcomes. The clinical manifestations of PAIS are characterized by acute encephalopathy, seizures, and central nervous system depression, and it carries significant long-term disabilities [37]. The true frequency of perinatal stroke is unknown, but various cases have been reported, in which the frequency ranged from 0.025% in live births [38] to 17% during autopsy of full-term newborns [39]. An imaging study of the brain is necessary to confirm a parenchymal hemorrhage with occlusion of the corresponding artery [40]. The route of possible cardiogenic emboli is shorter when passing through the aorta and the left carotid artery. That is why the left middle cerebral artery is the most frequent artery involved in stroke.

Studies suggest that multiple risk factors are involved in perinatal stroke and placental pathology may be a triggering factor [41]. Certain maternal and fetal conditions, including thrombotic vasculopathy and antiphospholipid syndrome, can lead to emboli entering the fetal circulation through the placental bloodstream.

Besides acquired contributing factors, the data suggests that genetic prothrombotic risk factors (the factor V (FV) G1691A mutation and the prothrombin (PT) G20210A variant) play a role in symptomatic neonatal stroke. Lipoprotein(a) elevated levels (>30 mg/dL) and protein C deficiency were also observed in neonates with PAIS [42].

Cerebral venous sinus (sinovenous) thrombosis (CSVT) is a rare but often underrecognized disorder with neurologic sequelae in up to 40% of survivors and a mortality approaching 10% [43]. Clinical manifestations of CSVT are similar to those of PAIS [44]. The neonate may experience sleep apnea or seizures (58%), diffuse neurologic signs (76%), focal neurologic signs (42%) [43], coma (30%), headache (18%), and motor weakness (21%) [45]. Rarely, anemia or thrombocytopenia can occur. Clinical symptoms may be nonspecific, which may obscure the diagnosis and delay treatment. Important manifestations of venous sinus thrombosis are papilledema and signs of increasing intracranial pressure. Vessels of the superior sagittal and transverse sinuses are most often involved in the pathological process. The most accurate method of diagnosis is contrast-enhanced MRI, but ultrasonogram is also used [46]. Mortality in venous sinus thrombosis has been estimated as from 2% to 24%, and the main complications are cerebral palsy, epilepsy and cognitive impairment [47]. Severe cases can lead to loss of limb function [48].

The development of neonatal intracardiac thrombosis is mainly associated with the installation of a central venous catheter (CVC), which is essential for the treatment of critically ill neonates. This type of thrombosis is associated with endocarditis development, pulmonary artery obstruction, ventricular dysfunction, and high mortality rate [49].

Intracardiac thrombosis often occurs in children after major surgical interventions associated with severe heart defects [50]. The study of newborns who underwent palliative treatment demonstrated that in 23–33% of cases, intracardiac thrombosis developed after the surgery [51]. Echocardiography is the most common diagnostic method with a minimally invasive strategy. The main clinical features in the newborn are the appearance of abnormal murmurs, persistent thrombocytopenia, and heart failure.

Renal vein thrombosis is the most common thrombosis that is not associated with the insertion of a catheter. The prevalence rate is 0.5 per 1000 admissions to the ICU. Unilateral renal vein thrombosis occurs in 70% of cases, and in 64% it localizes on the left side. The patients are mostly male [52]. The main clinical manifestations are macroscopic hematuria, palpable abdominal mass and thrombocytopenia. Other symptoms can be oliguria, proteinuria, acute renal damage, and increased blood pressure. The most common risk factors are prematurity and birth asphyxia. The main diagnostic method is a Doppler ultrasound. Echogenic clot, venous dilation, or lack of blood flow can be seen on the image. The most common complications of renal vein thrombosis are adrenal hemorrhage, inferior vena cava thrombosis, and hypertension.

The most common risk factors of portal vein thrombosis are omphalitis, sepsis and phototherapy in premature infants. Making a diagnosis can be challenging due to lack of clinical manifestations. In many children, portal vein thrombosis resolves on its own, but portal hypertension may manifest a decade after the neonatal period. A rather severe complication of thrombosis is cavernous transformation of the portal vein with splenomegaly and reverse portal blood flow [53].

In acute femoral artery thrombosis, there is usually a limb color change from pale to cyanotic. Femoral arterial pulse absence is an indication for ultrasound examination, which usually identifies the clot blocking the blood flow of the lower limb [54].

Neonatal purpura fulminans is a life-threatening condition that often develops a few hours after birth. This disorder manifests with an acute, rapidly progressive thrombosis of small-diameter vessels located primarily on the skin of the extremities. Purpura is characterized by the sudden development of intravascular thrombosis and hemorrhagic skin infarction, rapidly leading to disseminated intravascular coagulation (DIC) with consumption coagulopathy and shock symptoms. The first description of the disease dates back to 1962, but it was only several decades later that a link was established between purpura fulminans and a protein S deficiency [55]. There is a high mortality rate without immediate diagnosis and therapy intervention. The severity of clinical manifestation is dependent on the genetic variation of congenital proteins C and S deficiency. Skin lesions start out dark red, but eventually turn purple–black and necrotic. Spots often appear at trauma sites, such as where the intravenous catheter is inserted. A severe protein C deficiency is associated with retinal detachment, vitreous hemorrhage, and cerebral vascular thrombosis [56]. The following laboratory parameters are indicative of acute disease: thrombocytopenia, hypofibrinogenemia, increased fibrin degradation product levels, prothrombin time, and the activation time of blood platelet prolongation. Microangiopathic hemolytic anemia may also occur [57]. Neonatal fulminant purpura caused by congenital or acquired protein C or S deficiency remains a life-threatening condition. Early recognition of symptoms, fast-track diagnostics, and immediate substitution therapy can reduce the possible fatal outcome [58].

### 2.4. Risk Factors for Neonatal Thrombosis

Newborns are the most vulnerable patients to the development of thrombosis, and critically ill and premature infants have the highest risk. Neonatal thrombosis is usually secondary to underlying conditions or triggers, although a specific risk factor cannot always be identified. Risk factors for neonatal thrombosis are associated with the classic Virchow’s triad, which includes intravascular vessel wall damage, stasis of flow, and the presence of a hypercoagulable state. The development of neonatal thrombosis can be associated with both the health status of the mother and the neonate itself. Catheter-associated conditions are also singled out separately as risk factors.

Classification of risk factors for neonatal thrombosis (Table 2):(1)Catheter-associated risk factors for neonatal thrombosis.(2)Neonatal thrombosis risk factors associated with the neonatal condition.(3)Neonatal thrombosis risk factors associated with maternal condition.

#### 2.4.1. Catheter-Associated Risk Factors for Neonatal Thrombosis

The presence of a central venous catheter is one of the most frequent causes of thrombosis. About 60% of thrombotic episodes occur during the first year of life [59]. The venous catheter promotes clot formation by increasing protein absorption, platelets and leukocyte adhesion, and increased thrombin production. Another cause of thrombosis in children is catheter-associated infection, which occurs due to bacteria migration from the skin along the catheter. Studies demonstrated a positive association between thrombosis and infection in neonates (*p* < 0.05) [60]. Peripheral catheters are necessary for infants in the neonatal intensive care unit for drug administration, parenteral nutrition, and blood transfusion, which also present additional risk factors.

About half of all thrombotic events in neonates are arterial. Iatrogenic complications of in situ arterial devices such as umbilical-artery, peripheral-artery, or femoral-artery catheters are the main cause of clot formation. Most studies focus on umbilical venous catheters and peripherally inserted central catheters, and the available data on femoral venous catheters is limited [61]. Catheter insertion via the femoral vein in boys is an additional risk factor for neonatal thrombosis. Studies comparing the incidence of neonatal thrombosis due to central venous catheter placement (*p* = 0.01; OR = 8.2; 95% CI = 1.6 − 41.7) and femoral venous catheter placement show that femoral venous catheter insertion and use (*p* < 0.01) has a higher risk of neonatal thrombosis [62].

#### 2.4.2. Neonatal Thrombosis Risk Factors Associated with the Neonatal Condition

Coagulation system reduced compensatory mechanisms of neonates cause thrombotic complications. Studies show that healthcare-associated venous thromboembolism may occur due to artificial ventilation (OR = 7.27, 95% CI = 2.02 − 26.17, *p* = 0.002), infections (OR = 7.24, 95% CI = 2.66 − 19.72, *p* < 0.001), and extensive surgery (OR = 5.60, 95% CI = 1.82 − 17.22, *p* = 0.003) [63]. Neonatal thrombosis-associated risk factors include: gestational age, ICU length of stay, and congenital heart defects. Various infectious complications, such as systemic viral illness and sepsis, may also be risk factors for neonatal thrombosis development. The risk of thrombosis in neonates born after 37 weeks of pregnancy is significantly lower than in preterm infants. Some authors believe that the delivery option for childbirth, e.g., emergency C-section, may be important in neonatal thrombosis development. Risk factors for venous thromboembolism also includes preterm delivery (OR = 5.5, 95% CI 1.8 − 16.9) and a low Apgar score (OR = 9.2, 95% CI 1.9 − 45.2) [64].

Male sex (OR = 2.12) is an additional risk factor for neonatal thrombosis. The retrospective study with multivariate analysis suggests that respiratory distress syndrome is a significant predictor of neonatal thrombosis [65].

Thrombotic thrombocytopenic purpura (TTP), caused by ADAMTS-13 gene congenital mutation, can also provoke the microvascular thrombosis development of platelet consumption [66]. Plasma ADAMTS-13 deficiency increases the level of von Willebrand factor (vWF) multimers. The vWF/ADAMTS-13 axis dysfunction is considered as a risk factor in pediatric stroke and secondary microangiopathy development [66]. The vWF/ADAMTS-13 axis is involved in pathological clot formation, especially in neonates. Most studies suggest that neonates have lower ADAMTS-13 activity and higher levels of vWF than adults. However, like circulating APLs, a decrease in ADAMTS-13 level and an increase in vWF level in neonates does not directly cause thrombotic complications. Thrombosis may occur in combination with additional risk factors such as hypoxia, sepsis and other acute conditions, and the prolonged maintenance of central venous catheters.

#### 2.4.3. Neonatal Thrombosis Risk Factors Associated with Maternal Conditions

The main risk factor for neonatal thrombosis with respect to maternal conditions is the obstetric complication history, such as preeclampsia, placental insufficiency, and systemic inflammatory disorders. Preeclampsia is always associated with decreased placental blood flow due to placental abnormalities and vascular disorders of the placental bed. Maternal health conditions may be associated with hypercoagulation, which activates the coagulation cascade in the newborn [67]. For example, gestational diabetes mellitus causes vascular wall damage and endotheliopathy, which increases the risk of neonatal thrombosis [68]. Studies show that maternal risk factors include the presence of arterial hypertension (CI 1.05 − 1.91; *p* = 0.030) and thrombocytopenia (CI 1.59 − 3.06; *p* < 0.0001).

Neonatal thrombosis is also associated with inherited or acquired maternal or neonatal thrombophilia. The presence of maternal thrombophilia increases the coagulation potential and causes pre-thrombotic conditions during pregnancy. This mechanism leads to fetal placental thrombosis. The incidence of thrombotic complications is highest in the group of children born at 22–27 weeks of gestation [7].

### 2.5. Neonatal Thrombosis Management

In addition to the differences in the hemostasis system of newborns from adults, their pharmacological reactions to medications also differ. The development of convenient anticoagulants and the determination of doses based on body weight and age is important in evaluating treatment tactics for neonatal thrombosis. Currently, thrombolytics, heparin and, in critical cases, thrombectomy are used to manage neonatal thrombosis [69]. Treatment of catheter-associated neonatal thrombosis includes anticoagulants, low-molecular-weight heparin, thrombolytics, and symptomatic therapy [70].

Among numerous drugs used as anticoagulants in obstetric and pediatric practice, warfarin, UFH and low-molecular-weight heparin (LMWH) are the most well-known. Compared with warfarin and UFH, LMWH is less affected by nonspecific binding to plasma proteins and has increased bioavailability. The advantage of LMWH use is the binding reduction to platelet factor 4 and osteoblasts. This mechanism reduces the risk of heparin-induced thrombocytopenia and osteopenia. LMWH is the most commonly used anticoagulant in neonates. The risk of neonatal thrombosis recurrence after LMWH therapy is approximately 3%, based on the recent systematic review. Unfractionated heparin use is possible only with an adequate level of antithrombin. The therapy is started with an intravenous bolus injection of 50–100 units/kg, then the infusion is increased up to 400–500 units/kg.

The antithrombotic strategy includes thrombolytic therapy, thrombectomy and anticoagulants [69]. A therapeutic range is recommended for choosing the drug dosage to treat neonatal thrombosis. Some studies recommend the target anti-Xa range as 0.35–0.7 IU/mL, others set it at 0.5–1.0 anti-Xa IU/mL. For neonates on daily LMWH treatment, it is recommended that the drug dose is raised to a target range from 0.5 to 1 IU/mL.

Newborns with acute thrombosis are often administered fresh frozen plasma to help with clot reducing, as well as increasing the level of inhibitors of coagulation factors. Some thrombotic conditions that occur in neonates may not need antithrombotic medications, in contrast with older children and adults [71].

The limited warfarin use in pediatrics is associated with its shortened therapeutic index and increased risk of massive bleeding [72]. In addition, warfarin has many side effects and can interact with other medications, which certainly complicates its therapeutic use.

Direct oral anticoagulant use in children is still debated [73]. The classical anticoagulant treatment currently used in children has several limitations. Pharmacokinetics and pharmacodynamics are strongly associated with the age, clinical stability and other various factors. Direct oral anticoagulants should not be routinely used in children because we still lack information about appropriate dosage, safety, and efficacy. However, research on these drugs has great potential for the future.

### 2.6. Prevention of Neonatal Thrombosis

Neonatal thrombosis prevention in newborns is primarily aimed at eliminating the development of risk factors. Various genetic disorders of the hemostatic system, e.g., antiphospholipid syndrome and genetic thrombophilia in both parents should be diagnosed. Proper management of pregnancy is recommended in order to prevent placental insufficiency. Accurate diagnosis and treatment of maternal comorbidities help to prevent neonatal thrombosis. The prevention of catheter-associated neonatal thrombosis with anticoagulant medication use has no evidence base [59].

Perioperative antithrombotic prophylaxis in adults is widely recommended, but there are no clinical trials that evaluate the effectiveness and safety of antithrombotic therapy in children. Generally accepted protocols for perioperative prophylaxis also do not exist [74].

Preventing neonatal thrombosis by preventing risk factors in a timely and appropriate manner will improve the treatment options for newborn children.

### 2.7. COVID-19 and Neonatal Thrombosis

Coronavirus infection is a risk factor for thrombosis, especially in patients with severe disorders. Patients with COVID-19 infection suffer from disproportionately activated complement as well as excessive coagulation, which leads to thrombotic complications and unfavorable disease outcome [75]. COVID-19 disease predisposes patients to more frequent thrombotic events. Currently, little is known about COVID-19’s effect on the placenta and the newborn.

COVID-19 is associated with the multisystem inflammatory syndrome in children, more often in the late stages of the disease, when the causative agent is no longer detectable by PCR. The clinical manifestation varies, but often presents with Kawasaki-like symptoms such as febrile fever, conjunctivitis, polymorphic maculopapular rash, frequent gastrointestinal symptoms, elevated levels of C-reactive protein, liver transaminase, procalcitonin, lactate dehydrogenase, creatine phosphate kinase, interleukin-6 and interleukin-10 [76].

Newborns with COVID-19 infection have a wide range of clinical manifestations, from asymptomatic carriage to critical condition. Neonatal cerebral venous sinus thrombosis causes seizures and results in high neonatal mortality. Fetal umbilical artery thrombosis is a rare condition caused by hereditary thrombophilia, hypoxia, dehydration and placental dysfunction. COVID-19 is associated with coagulopathy and thromboembolism in adults and children, which leads to preterm labor, fetal growth restriction, miscarriage, and impaired placental perfusion [77]. Neonatal COVID-19 infection has horizontal transmission in the postnatal period.

In children, COVID-associated hemostatic abnormalities result in the prolongation of PT and activation time of blood platelets, and increased D-dimer and fibrinogen levels. It should be mentioned that D-dimer levels measured during the neonatal period are found to be higher than adult values. Therefore, the increased D-dimer-value concentration in COVID-19 newborn infants should be interpreted with caution. COVID-19-related proinflammatory cytokines induce endothelial damage, resulting in primary hemostasis activation and tissue factor overexpression [78]. Increased platelet procoagulant activity combines with plasmin activity inhibition due to decreased levels of urokinase-type plasminogen activator and increased levels of plasminogen activator inhibitor-1. This mechanism promotes fibrin deposition, forming local microthrombi and worsening clinical outcomes. Mild thrombocytopenia or thrombocytosis may also occur. Elevated D-dimer levels in patients with COVID-19 are associated with greatly increased thrombin generation, secondary to endothelial activation. Activation is induced by infectious stimulation, severe hypoxemia, and local pulmonary microthrombosis. A multicenter retrospective cohort study reported cases of venous thromboembolism in 2.1% of patients with COVID-19 and 6.5% of patients with multisystem inflammatory syndrome [79]. Thrombotic complications include deep veins, pulmonary vein and cerebral venous sinus thrombosis.

Cerebral venous thrombosis is a rare but dangerous complication of COVID-19 that occurs in both adults and children. Cerebral venous thrombosis occurs in preterm infants whose mother became infected with the coronavirus infection in the third trimester of pregnancy. Severe neonatal thrombosis in these clinical cases develops as a result of maternal COVID-19 infection, which affects the neonate’s prothrombotic status [80]. Several clinical studies demonstrate fetal vascular malperfusion or fetal thrombotic vasculopathy in COVID-19-positive women in 10 out of 20 cases [81]. Maternal COVID-19 infection can also cause fetus pathology including medically preterm delivery, growth restriction, and miscarriage [82].

Early diagnosis of neonatal central venous thrombosis is complicated due to the rarity of conditions and variable clinical manifestations. With maternal COVID-19 infection in the first trimester, an additional mid-trimester ultrasound scan is recommended. If the infection occurs in the second or third trimester, additional third-trimester growth assessment is recommended. Coronavirus infection is not an indication for preterm delivery or C-section, and the timing of delivery should be determined by the severity of the illness, maternal comorbidities, gestational age, and the “mother–fetus” condition [83].

The high incidence rate of COVID-associated thrombosis demands appropriate thromboprophylaxis in newborns. Anticoagulant therapy with low-molecular-weight heparin or unfractionated heparin for subcutaneous administration is recommended, especially in children and newborns with risk factors. In patients with multisystem inflammatory syndrome, aspirin administration is also recommended until laboratory parameters recover.

COVID-19 is associated with a high incidence of preterm delivery and preeclampsia, compared to pregnancies without COVID-19. Thrombophilic status abnormalities caused by maternal coronavirus infection can increase the risk of the neonatal venous thrombosis development, which requires serious clinical investigation and early diagnosis. Further studies are needed to clarify the pathophysiological mechanisms of COVID-19 during pregnancy and the COVID-19 impact on fetal development.

## 3. Conclusions

Neonatal thrombosis is an extremely dangerous condition for newborns. The pathogenesis of neonatal thrombosis is complex, and coagulation abnormalities in the neonate are a challenge for caregivers. The hemostasis system is different from that of adults, so the physician needs to understand the pathogenesis of the disease in children. The clinical manifestations of neonatal thrombosis are very diverse, and vary from the absence of any symptoms to the appearance of life-threatening conditions. Thrombosis development in the newborn is caused by a combination of different risk factors in both the child and the mother. The COVID-19 pandemic has also adjusted the pathogenesis and clinical manifestations of neonatal thrombosis. Coronavirus infection is a risk factor for coagulopathy and thromboembolism development and causes adverse outcomes.

In future investigations, it would be interesting to conduct long-term observations of children in order to identify long-term consequences of patients with neonatal thrombosis in their past. The diagnosis and treatment of neonatal thrombosis need to be standardized, taking into account more prospective clinical studies. In children, the diagnosis of blood clotting disorders is based not only on the medical history, but also on the family history. From this point of view, testing for genetic blood clotting disorders can benefit both parents.

Generally, testing for coagulation abnormalities is indicated in all sick newborns. But it is important to emphasize that the standard and more common procedure for taking a capillary blood sample via a heel lance allows for the obtaining of only a small amount of blood for laboratory analysis and cannot be used to obtain a large sample volume for specific testing, such as a clotting profile. Sampling from arteries and veins is not always feasible in newborns, and the risks associated with indwelling catheters such as thrombosis and infection limit the duration that they can be left in situ. For example, light transmission platelet aggregometer (LTA), a standard technique used to monitor platelet function, requires a large amount of blood and takes a lot of time. There is growing need for bed-side assays for easier and faster diagnostics of coagulation disorders. From this point of view, the Impact-R platelet analyzer has some advantages because it uses a small amount of whole blood (0.13 mL) rather than platelet rich plasma, no centrifugation step, and measures platelet adhesion and aggregation under physiological conditions such as shear stress and presence of a thrombogenic surface.

Currently, no unambiguous and unidirectional prevention strategies of neonatal thrombosis have been developed; therefore, adequate and correct pregnancy control and reduction of risk factors is recommended. Anticoagulant medication has many side effects, but its use is necessary for effective treatment. Careful therapy choices include the pharmacodynamics and the pathophysiology of neonatal thrombosis data. It is possible that the high rate of thrombosis can be attributed to better diagnostic quality and increased monitoring during pregnancy.

## Figures and Tables

**Figure 1 ijms-24-13864-f001:**
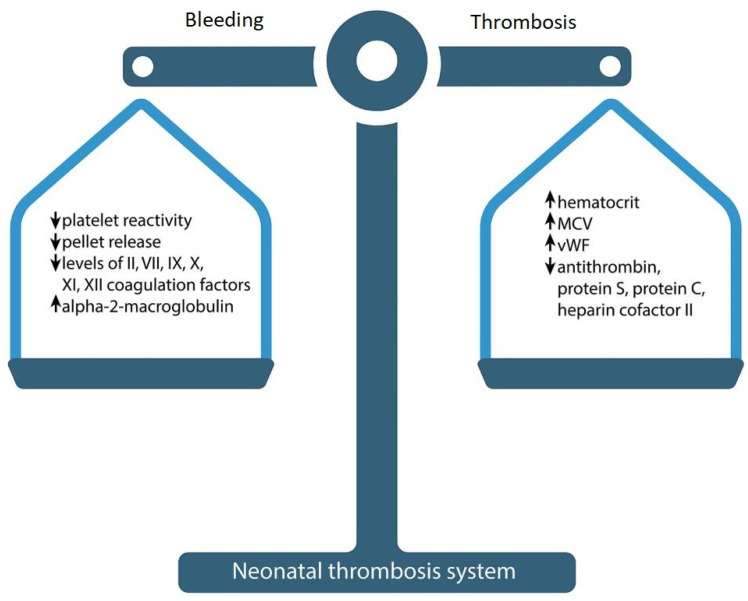
Neonatal hemostasis.

**Table 1 ijms-24-13864-t001:** Coagulation factors and natural anticoagulant levels changes in full-term neonates at term and at 6 months.

Hemostasis System Parameter	At Birth	6 Months
Antithrombin	40–60%	adult level at day 90
Protein S	40–60%	adult level at day 90
Protein C	low	adult level not reached
VWF	mean 153%	drops to ≈ 100%
Fibrinogen	adult level	adult level
Vitamin K-dependent factors (II, VII, IX, X)	mean ≈ 40–50%	80–90% of adult level
Contact system factors (factors XI, XII, prekallikrein, HMWK)	mean ≈ 40–50%	80–90% of adult level
Factor V and XIII (both a- and b-unit)	mean 70–80%	adult levels at day 5
FVIII	mean 100%	drops slowly to ≈ 75%
Factor V and XIII	mean 70–80%	adult levels at day 5
a2-macroglobulin	high	increasing further

**Table 2 ijms-24-13864-t002:** Classification of risk factors for neonatal thrombosis.

Catheter-Associated Risk Factors for Neonatal Thrombosis	Risk Factors Associated with the Neonatal Condition	Risk Factors Associated with Maternal Condition
Central venous catheter Parenteral nutrition Blood transfusion Intravenous medication administration Catheter-associated infections Femoral venous catheter	Lung Ventilation Systemic viral infections and complications Gestational age ICU length of stay Congenital heart defects Sepsis Emergency C-section Preterm birth Low Apgar score Malignant neoplasms in the neonate Male sex Acute respiratory distress syndrome	Preeclampsia Placental Insufficiency Systemic inflammatory disorders Gestational diabetes mellitus Hereditary or acquired thrombophilia Hypertension Thrombocytopenia Low ADAMTS-13 activity High vWF level

## Data Availability

Not applicable.

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
