# Peer review of "The Hemostatic System in Newborns and the Risk of Neonatal Thrombosis"

_ijms, 2023, doi:10.3390/ijms241813864_

Round 1

Reviewer 1 Report

A well written and structured review of an important clinical topic, often ignored in the scientific literature. The authors successfully and succinctly provide an overview of this topic in a comprehensive manner- well supported by the relevant literature. Only minor comments to make-

1. While well written, there are small nuanced grammatical and syntax errors through out. I would recommend a proof read and edit by a native English speaker to catch these. 

2. Table 2 requires reorganisation.

3. I think the authors should allude to testing methodologies and technologies, and how this has improved the diagnosis and treatment of neonatal thrombosis and associated conditions e.g. LTA versus IMACT-R etc., due to limited volumes of samples.

See above, I would recommend a proof reading service. It is a well written and structured review, but subtle errors through out that could be improved upon.

Author Response

Dear Reviewer, 

Thank you very much for taking the time to review this manuscript. 

Please find the responses below. All corrections and additions are highlighted in red.

  1. English grammar was checked and corrected by native speaker 
  2. Table 2 was reorganized.
  3. We added a paragraph regarding different testing methodologies (page 13, lines 580-593) ... 

"The diagnosis and treatment of neonatal thrombosis need to be standardized taking into account more prospective clinical studies. Generally, testing for coagulation abnormalities is indicated in all sick newborns. But it is important to emphasize that the standard and more common procedure for taking a capillary blood sample via a heel lance allows to obtain only a small amount of blood for laboratory analysis and cannot be used to obtain a large sample volume for specific testing, such as a clotting profile. Sampling from arteries and veins is not always feasible in newborns, and the risks associated with indwelling catheters such as thrombosis and infection limit the duration that they can be left in situ. For example, light transmission platelet aggregometer (LTA), a standard technique used to monitor platelet function, requires a large amount of blood and takes a lot of time. There is growing need for bed-side assays for easier and faster diagnostics of coagulation disorders. From this point of view, the Impact-R platelet analyzer has some advantages because uses small amount of whole blood (0,13 ml) rather that platelet reach plasma, no centrifugation step and measures platelet adhesion and aggregation under physiological conditions such as shear stress and presence of a thrombogenic surface…"

Yours Sincerely, 

Jamilya Khizroeva

Reviewer 2 Report

The authors have produced an excellent review on the particularities of coagulation in newborns and the risk of neonatal thrombosis. I simply suggest adding a short paragraph on the biological investigations that can be proposed in the case of neonatal thrombosis, depending on the clinical signs, and keeping in mind the difficulty of sampling and the low volume of blood that can be taken in a newborn. Can the search for a coagulation abnormaly in both parents be a prerequisite?

- In the table 1, the authors indicate (and it is true) that factors V and VIII levels are almost  in the normal range at birth, but in the Fig 1 they indicate that FVactor V is decreased. The figure should be in agreement with the table.

- The authors mention heparin cofactor II as a physiological inhibitor of coagulation. Although it is true that it acts by inhibiting thrombin, there is no convincing evidence (except some case reports)  that HCII deficiency is related to thrombotic events. This should be underlined.

- page 5, the authors explain that soluble thrombomodulin is increased at birth in asphyxiated full-term infants. Soluble thrombomoulin acts as an anticoagulant, but it could be interesting to explain, in this review, why it is an anticoagulant. 

- page 11, in the paragraph cncerning COVID-19, the authors indicates that elevated d-dimer levels in patients with COVID-19 associates with thrombin generation...", line 493. It should be mentionned in the text that D-dimer are elevated in newborns, therefore the utility of their measurement in newborn is low.

- page 2, line 50: it is alpha-2-antiplasmin but not alpha-1....

Author Response

Dear Reviewer!

Thank you so much for your comments.  We tried to take into account all your suggestion and comments in updated manuscript. All changes are highlighted in blue.

1) Thank you for pointing this out. We modified the fig 1. 

2) We underlined the role of heparin cofactor II (page 5, lines 173-181):

..."The role of heparin cofactor II (so called “minor” antithrombin) in the modulation of neonatal hemostasis system is still has been questioned. HCII selectively inhibits thrombin and has a moderate binding affinity to heparin and dermatan sulfate. The rate of thrombin inhibition by HCII is significantly slower that by antithrombin and the plasma level of HCII is 25-50% that of AT. Term infants and healthy preterm newborns have lower level of HCII which more likely reflects immature liver function and attains adult levels at 5 to 7 months of life. Neither adult and children with thrombosis history had significantly lower levels of HCII activity. Considering that the absence of HCII does not strongly correlate with thrombosis, its physiological functions are still under investigation."...

3) We’ve accordingly added the paragraph concerning the role of TM as anticoagulant (page 5, lines 184-195).

"Thrombomodulin (TM) is a multidomain glycoprotein receptor for thrombin and is best known for its role as a cofactor in a clinically important natural anticoagulant pathway of protein C. It binds thrombin and converts it from a procoagulant to an anticoagulant enzyme that activates protein C. TM is expressed at the surface of endothelial cells and exists in a soluble form (sTM) in plasma and urine. Loss of TM from the surface of endothelial cells may favor thrombosis. TM have not only anticoagulant activity but also exerts anti-inflammatory properties using APC-dependent and APC-independent mechanisms. In addition to its anticoagulant and anti-inflammatory function, the TM–thrombin complex activates thrombin‐activatable fibrinolysis inhibitor (TAFI), which inhibits tissue plasminogen activator (tPA)-induced fibrinolysis (antifibrinolytic properties of TM). Inflammation has been shown to reduce TM expression on the endothelial surface and this decrease may contribute to the hypercoagulable condition which is characteristic for inflammatory states"

4)  Also, we have added the information about the D-dimer in newborns (page 12, lines 508-511)

"It should be mentioned that D-dimer levels measured during the neonatal period are found to be higher than adult values. Therefore, the increased D-dimer values concentration in COVID19 newborn infants should be interpreted with caution"...

5) We fixed antiplasmin 1 by 2 (page 2)

6) We added a short paragraph on the future investigations (page 12, lines 551-553; page 13, lines 561-574)

The pathogenesis of neonatal thrombosis is complex and coagulation abnormalities in the neonate are a challenge to caregivers. In future investigations, it would be interesting to conduct long-term observations of children in order to identify long-term consequences of patients with neonatal thrombosis in their past.

The diagnosis and treatment of neonatal thrombosis need to be standardized taking into account more prospective clinical studies. Generally, testing for coagulation abnormalities is indicated in all sick newborns. But it is important to emphasize that the standard and more common procedure for taking a capillary blood sample via a heel lance allows to obtain only a small amount of blood for laboratory analysis and cannot be used to obtain a large sample volume for specific testing, such as a clotting profile. Sampling from arteries and veins is not always feasible in newborns, and the risks associated with indwelling catheters such as thrombosis and infection limit the duration that they can be left in situ. In children, the diagnosis of blood clotting disorders is based not only on the medical history, but also on the family history. From this point of view, testing for genetic blood clotting disorders can benefit both parents.

Yours Sincerely, 

Corresponding author Jamilya Khizroeva
